# Effects of Afforestation on Plant Diversity and Soil Quality in Semiarid SE Spain

Carmen Segura [1], María N. Jiménez [2], Emilia Fernández-Ondoño [3] and Francisco B. Navarro [4],*

1   Department of Sustainable Agriculture Sciences, Rothamsted Research, North Wyke, Okehampton EX20 2SB, UK; carmen.segura-quirante@rothamsted.ac.uk
2   Department of Botany, Faculty of Pharmacy, Campus de Cartuja, University of Granada, 18071 Granada, Spain; mnoelia@ugr.es
3   Department of Soil Science and Agricultural Chemistry, Faculty of Science, University of Granada, C/Severo Ochoa, s/n, 18071 Granada, Spain; efernand@ugr.es
4   Area of Agriculture and Environment, Institute of Agricultural Research and Training of Andalusia (IFAPA), Government of Andalusia, Camino de Purchil s/n, 18004 Granada, Spain
*   Correspondence: fbruno.navarro@juntadeandalucia.es

**Abstract:** Farmland afforestation has been promoted in recent decades and is one of the main strategies included in the UN Decade on Ecosystem Restoration to recover degraded areas. However, the impacts of afforestation on plant diversity and soil quality indicators are still not well-understood in semiarid environments. In this study, we assessed the relationships between plant diversity indicators (abundance, total richness, richness by functional groups, and Shannon diversity) and a large number of variables in 48 afforestation sites in southeast Spain. We considered associated environmental factors, such as geographical, climatic or edaphic variables, age, and land-use history. We compared plant diversity and soil properties following land-use change from cereal cropping to afforestation, which is one of the most common land-use changes in Mediterranean areas. Plant diversity in afforested sites was found to be dependent on previous land use, the proximity of natural vegetation, several soil properties (texture, pH, and total nitrogen), and plantation age. Afforested soils showed higher plant diversity and an improvement in edaphic parameters related to multifunctionality in semiarid ecosystems (i.e., soil organic carbon, nitrogen, and potassium) than arable cropped soils.

**Keywords:** active restoration; functional groups; plant richness; soil properties; land-use change; Mediterranean region; UN Decade on Ecosystem Restoration

## 1. Introduction

In recent decades, various global and regional initiatives have promoted active environmental restoration as one of the main worldwide nature-based solutions to prevent soil erosion, recover degraded lands, and limit the effects of climate change [1]. In this sense, the area that has been reforested (planting trees in previously forested landscapes) and afforested (planting trees in landscapes that were not previously forested) has increased by over 123 million ha since 1990, and currently represent 7% of the world's forest area [2]. Scant efforts have been devoted to monitoring the success of tree planting in achieving objectives and assessing the success of restoration [3]. The United Nations General Assembly declared 2021–2030 as the Decade on Ecosystem Restoration, with an aim of planting billions of trees through restoration initiatives around the world [3]. This strategy is derived from the Sustainable Development Goals Agenda (i.e., Sustainable Development Goal 15.3), addressing both climate change mitigation and biodiversity and ecosystem services [4].

This new emphasis on tree planting is occurring within a context of intense debate regarding the suitability of adopting passive or active restoration strategies. The relative effectiveness of active restoration strategies relative to passive recovery of former agricultural land for ecosystem functions is not well-understood [5]. It has been proposed that the active afforestation of degraded fields or agricultural areas should only be





adopted when ecosystem restoration may be impossible passively [6–8]. For some authors, afforestation is often the only viable approach to prevent soil erosion and restore ecosystem functions, and others promote active intervention as an opportunity for climate change mitigation that could have positive effects on biodiversity, particularly when native species are planted [9–11].

The influence of afforestation on plant diversity and soil quality is dependent on land-use history, whether tree species are planted, the time elapsed since restoration began, and landscape context, amongst others environmental variables [6,12–15]. Studies conducted under dry conditions that incorporate afforestation success considering both plant diversity and soil quality recovery are scarce, even though semiarid and subhumid areas account for 25% of the terrestrial Earth, and a huge agricultural area (cultivated or abandoned) has been afforested in recent decades. These regions are considered extremely vulnerable to land degradation and desertification [2,10,16]. Previous land use is a key driver of plant diversity and soil quality changes following afforestation [5,17]. However, some studies conducted in semiarid that included both plant diversity and soil have focused on grasslands and natural vegetation instead of farmland afforestation [18,19]. Other studies that reported positive links between plant diversity and edaphic properties, such as soil organic carbon (SOC), total nitrogen (TN), phosphorus (P), and potassium (K), have been conducted in secondary forests, abandoned lands, and aromatic species cultivation [20–22].

Studies that report soil quality improvement following afforestation of arable land did not consider plant diversity [23,24]. Abandonment of agricultural land presents opportunity for the recovery of soil properties related to fertility, and an increase in biodiversity with little need for active restoration and at less expense [25]. Understanding which edaphic and other environmental variables explain plant diversity under semiarid conditions, particularly in afforestation projects, could support any decision-making processes. However, despite soil being recognized as a main driver of plant diversity, the extent to which edaphic parameters control plant diversity is still not well-known [26].

The uncertainties related to the effects of afforestation on plant diversity and soil have important implications for European Union Common Agricultural Policy subsidies [27–29]. European afforestation efforts have been included in the Common Agricultural Policy since 1992, mainly in Mediterranean countries such as Spain, Portugal, and Italy [30]. Under this framework, the Farmland Afforestation Scheme aimed to reduce product surpluses, promote early retirement opportunities for farmers, increase forest resources, and provide soil and biodiversity protection by offering farmers annual incentive payments for the conversion of farmland to afforestation [31]. More than 730,000 ha of farmland has been afforested in Spain, approximately 50% of which was previously devoted to arable crops [32]. The lack of planning and measurable objectives has resulted in afforested sites spread haphazardly, not following any technical or territorial criteria. This has complicated the wide-scale monitoring of afforestation impacts on biodiversity and soil quality, especially when baseline assessment has not been established [33,34].

With an anticipated expansion of afforested land by the end of the 21st century, it is important to ensure not only the accurate management of existing afforestation, but also the careful planning of newly planted forests to maximize the benefits for biodiversity and sustainability [1,3,35]. An accurate assessment of soil change following afforestation should also be included as a principal component of restoration monitoring [36,37], even though many factors are involved and relationships between plant diversity and soil may not be evident [26,38,39]. Identifying environmental factors that indicate afforestation success as well as quantifying changes in plant diversity and soil quality indicators following planting will help improve restoration success.

In this study, we aimed to (i) elucidate which environmental variables can explain plant diversity, and (ii) compare plant diversity and edaphic properties in paired cereal crops and afforestation sites. We focused on plant species richness, plant diversity, and soil quality indicators in a typical semiarid Mediterranean area. Species richness and diversity are among the principal metrics chosen to assess the success of restoration initiatives from

a biodiversity point of view: they are drivers and predictors of ecosystem multifunctionality (understood as the ability of ecosystems to provide and maintain multiple functions and services simultaneously), for instance, by buffering the effects of climate change and desertification in semiarid climates, and by influencing nutrient cycling and carbon storage [40,41]. Some recommendations for evaluating plant diversity restoration have encouraged the assessment of functional groups [36,38]. Identification of endemism levels and conservation status is crucial to the evaluating of the active restoration effects on biodiversity [12], especially in biodiversity hotspots such as the Mediterranean region.

We hypothesized that (1) previous land use, location, climate, and edaphic properties, such as texture, SOC, macronutrients, and parameters related to soil water content, can explain plant diversity in Mediterranean semiarid afforestation; and (2) afforestation improves plant diversity and soil quality indicators, compared to cereal cropping in semiarid environments. We explored environmental variables that could explain plant diversity in afforested farmlands using generalized linear models, quantifying and comparing plant diversity and soil quality indicators in paired afforestation and cereal crop sites. To solve the lack of baseline sampling before afforesting, we selected cereal crops as control sites because half of the afforested area in the study region was arable land [6,27,32].

## 2. Materials and Methods

### 2.1. Study Area

The study area lies in the northeast of the Province of Granada (southeast Iberian Peninsula) within the confines of the Baetic mountain system (Figure 1). It extends over 5220 km$^2$ and includes the administrative regions of Guadix, Baza, and Huéscar. Within this area, a total of 241 farms were afforested between 1993 and 2006. The size of the farms varied between 1.15 and 97.73 ha (average of 26.85 ha $\pm$ 1.36 SE). The most common tree species were *Pinus halepensis* Mill. planted in single or mixed stands with *Quercus ilex* L. subsp. *ballota* (Desf.) Samp. These were planted at low densities (300–500 stem ha$^{-1}$) using the linear subsoiling technique. No silvicultural treatments or grazing were adopted in the afforested farmlands over time. We selected a total of 48 of these afforested farms for this study according to criteria of representativeness of the three administrative regions, accessibility, and economic cost relative to sampling effort and soil analysis (Figure 1).

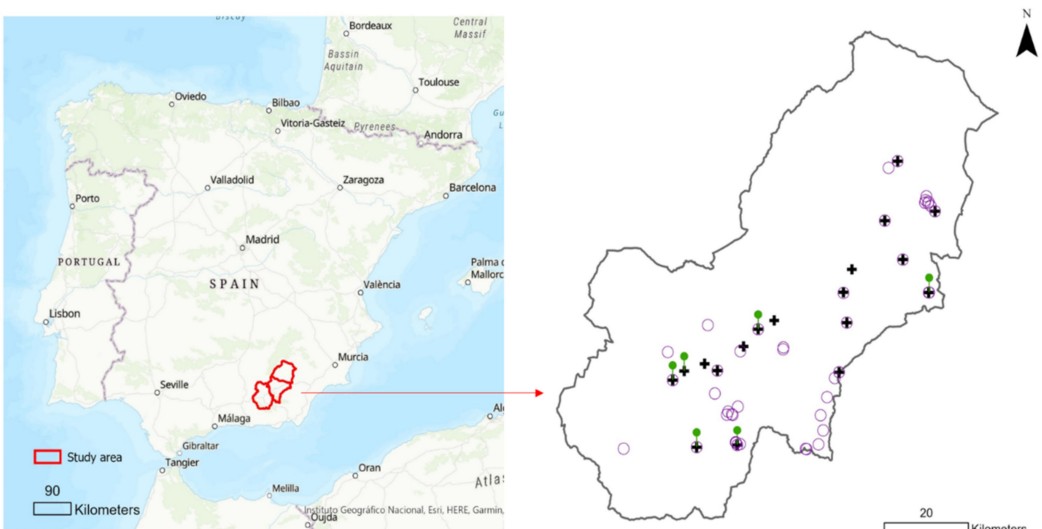

**Figure 1.** Study area and sites included. Purple circles indicate the location of afforested sites sampled in 2006 (N = 43). Green pins indicate the location of paired sites sampled in 2006–2007 to perform the diversity comparison between afforestation (N = 6) and cereal crops (N = 6). Black crosses indicate the location of paired sites, afforestation (N = 18), and cereal crops (N = 18) where soil properties were measured in 2011.

The region has a Mediterranean macroclimate within meso-, supra-, and oromediterranean thermotypes and semiarid, dry, and subhumid ombrotypes [42]. The most frequent soils are Leptosols (Lithic Leptosols and Rendzic Leptosols), Calcisols (Petric Calcisols and Haplic Calcisols), Cambisols (Eutric Cambisols and Calcaric Cambisols), and Regosols (Leptic Regosols, Calcaric Regosols and Eutric Regosols). There are also small areas of Calcaric Phaeozems [43].

### 2.2. Data Collection

From the 48 farmlands afforestation, we selected 43 sites for the study of explanatory environmental variables for plant diversity (Table 1 and Figure 1; Supplementary Materials Table S1). Within each site, a sample plot (20 × 20 m) was identified, at least 50 m away from the edge to avoid any edge effects. For each plot, five 20 m long linear transects were selected, separated from each other by 4 m, and aligned perpendicular to furrows made during planting.

**Table 1.** Environmental data collected for each farmland afforestation (N = 43). * Nominal variables.

| Variable Type | Variable | Unit | Mean (SD) | Minimum | Maximum |
|---|---|---|---|---|---|
| Geographic | Elevation above sea level (Elevation) | m | 1237.0 (304.2) | 881.0 | 1939.0 |
| | Distance to crops (D_crops) | m | 188.5 (308.0) | 0.0 | 1488.0 |
| | Distance to natural vegetation (D_n_veg) | m | 61.9 (133.7) | 0.0 | 531.5 |
| | Distance to shrub patches (D_shrub) | m | 1241.0 (1022.0) | 90.5 | 4675.0 |
| | Distance to *Quercus* woodland (D_Qil) | m | 1727.0 (2219.0) | 0.0 | 9834.0 |
| Climatic | Annual Precipitation (Pp) | mm | 394.0 (65.2) | 311.0 | 547.6 |
| | Mean Temperature (T) | °C | 12.68 (0.98) | 10.90 | 14.10 |
| Edaphic | Available water (AW) | % | 6.82 (2.90) | 2.22 | 12.74 |
| | Soil moisture at field capacity (33 kPa) (SM33) | % | 16.24 (5.78) | 5.09 | 28.58 |
| | Soil moisture of the wilting point (1500 kPa) (SM1500) | % | 9.42 (3.81) | 2.87 | 19.02 |
| | Water saturation (WS) | % | 38.39 (7.59) | 27.63 | 56.42 |
| | Gravel (>2 mm) | % | 37.24 (18.02) | 0.00 | 73.42 |
| | Sand (2–0.05 mm) | % | 45.28 (16.75) | 3.23 | 75.65 |
| | Coarse silt (0.05–0.02 mm) (CSilt) | % | 10.27 (4.35) | 3.07 | 24.67 |
| | Fine silt (0.02–0.002 mm) (FSilt) | % | 22.15 (9.66) | 8.37 | 57.25 |
| | Clay (<0.002 mm) | % | 22.30 (8.99) | 6.12 | 38.47 |
| | Cation exchange capacity (CEC) | $Cmol_{(+)}kg^{-1}$ | 11.28 (6.13) | 3.27 | 25.39 |
| | Electrical conductivity (EC) | $mS\,cm^{-1}$ | 0.89 (0.40) | 0.51 | 2.21 |
| | Exchangeable sodium (Na) | $Cmol_{(+)}kg^{-1}$ | 0.06 (0.07) | 0 | 0.29 |
| | Exchangeable magnesium (Mg) | $Cmol_{(+)}kg^{-1}$ | 1.86 (1.28) | 0.30 | 5.98 |
| | Exchangeable potassium (K) | $Cmol_{(+)}kg^{-1}$ | 0.36 (0.27) | 0.06 | 1.28 |
| | Soluble calcium (s_Ca) | $mg\,L^{-1}$ | 3.71 (2.41) | 0.67 | 9.81 |
| | Soluble magnesium (s_Mg) | $mg\,L^{-1}$ | 0.65 (0.47) | 0.16 | 2.59 |
| | Soluble potassium (s_K) | $mg\,L^{-1}$ | 0.40 (0.51) | 0.02 | 3.05 |
| | Soluble sodium (s_Na) | $mg\,L^{-1}$ | 1.61 (0.98) | 0.13 | 4.78 |
| | CaCO$_3$ (CaCO3) | % | 24.36 (23.24) | 0.10 | 73.78 |
| | pH | | 7.90 (0.55) | 6.72 | 8.83 |
| | Soil organic carbon (SOC) | % | 1.24 (0.82) | 0.30 | 3.19 |
| | Total nitrogen (TN) | % | 0.11 (0.05) | 0.05 | 0.24 |
| | $NO_2^-$ (NO2) | $mg\,L^{-1}$ | 3.29 (8.65) | 0.00 | 41.60 |
| | $NO_3^-$ (NO3) | $mg\,L^{-1}$ | 11.10 (44.41) | 0.00 | 233.40 |
| | $SO_4^{2-}$ (SO4) | $mg\,L^{-1}$ | 46.60 (64.58) | 6.81 | 301.50 |
| | $Cl^-$ (Cl) | $mg\,L^{-1}$ | 30.25 (22.01) | 10.53 | 117.2 |
| Geological | Lithology | * | | | |
| Land use | Age of afforestation | years | 8.4 (2.1) | 5.0 | 12.0 |
| | Previous land use (cereal crop or old-field) | * | | | |
| | Plant cover (Cover) | % | 67.49 (13.15) | 40.00 | 91.00 |
| | Cover of *Pinus halepensis* (CovPha) | % | 5.93 (6.44) | 0.00 | 32.00 |
| | Cover of *Quercus ilex* subsp. *ballota* (CovQil) | % | 0.26 (0.62) | 0.00 | 3.00 |

The data of vascular plants (including afforested species) were recorded where the tip of a pointer touched the vegetation perpendicularly every 100 cm along transects, in a manner similar to the point quadrat technique (cf. [44]). We estimated species abundance (number of individuals of each species per plot) and relative cover (%) of the aerial parts of each species. Bare soil (%) was also estimated. All species were identified and classified into functional groups: annuals (including annual or biennial plants), perennial forbs, perennial grasses, and woody species (including dwarf scrubs and shrubs). Conservation status was checked to quantify endemism and to identify endangered species, according to Cabezudo et al. [45].

Plant diversity was considered in terms of species richness and species diversity. Species richness is reported as the total number of species per plot (Total_R) and as richness by functional group (Annuals_R, Forbs_R, Grasses_R, and Woody_R). Species diversity was calculated as the Shannon–Wiener index (H′):

$$H' = -\Sigma p_i \ln p_i, \tag{1}$$

where $p_i$ is the abundance of i expressed as a proportion of the total species abundance in each afforested plot.

Differences in plant diversity both including and excluding afforested species were examined (Total_H and H_noaff, respectively).

The edaphic characteristics and geographic, climatic, geological, and land use variables for each afforested site are summarized in Table 1. Composite soil samples were collected from the center and corners of each quadrat to a depth of 0–10 cm and mixed to obtain one representative sample per plot. Soil samples were air-dried and sieved (<2 mm). Gravels (>2 mm) were weighed and stored separately. Soil texture was analyzed by the Robinson pipette method [46]. Available water content was calculated by the difference between moisture content at field capacity extracted in a pressure plate at −33 kPa and moisture at plant wilting point, measured at −1500 kPa [47]. Exchangeable bases ($Mg^{2+}$, $K^+$, and $Na^+$) were extracted with 1 N $NH_4OAc$, and cation-exchange capacity was determined by saturation in sodium by washing the soil samples with alcohol and extracting the sodium adsorbed with 1 N $NH_4OAc$ [46]. pH was measured in a soil suspension in distilled water (1:2.5). Soil organic carbon was determined using the Walkley and Black method [48], modified by Tyurin [49]. The Kjeldahl method was used to calculate total N [50]. $CaCO_3$ equivalents were determined using the manometric method [51]. A saturated extract was prepared from each sample to determine its electrical conductivity [52]. Soluble calcium, magnesium, potassium, and sodium in the saturated extracts were determined by atomic-absorption spectrometry. Available phosphorus (only measured in 2011) was determined using Olsen's method [53]. $NO_2^-$, $NO_3^-$, $SO_4^{2-}$, and $Cl^-$ in the saturated extracts were determined by high-precision liquid chromatography (HPLC).

Geographic variables were obtained from the Land Use and Vegetation Cover Map of Andalusia (scale 1:25,000) [54]. Climate data were derived from information provided by the National Meteorological Institute of Spain using the interpolation method proposed by Sánchez-Palomares et al. [55]. Lithological data were obtained from geological maps published by the Spanish Institute for Geology and Mining [56]. Lithology was divided into the following categories: (1) carbonate rocks, including limestones, dolomites, marbles, limestone crusts, and conglomerates; (2) siliceous rocks, including mica schists, phyllites, and quartzites; (3) silts with clay; and (4) sands.

Data relating to plantation age, land-use and management were provided by the Ministry of Agriculture and Fisheries of Andalusia. Relative covers of *P. halepensis* and *Q. ilex* subsp. *ballota* measured within the sampled afforested areas were included as land-use variables.

A paired site comparison approach was followed to compare plant diversity and soil characteristics between afforested and cereal crop sites because baseline sampling before afforesting was unavailable. A total of six paired sites with similar ecological characteristics were chosen to compare plant diversity (Figure 1). Additional soil sampling was carried out

in 18 paired sites to compare soil parameters at two soil depths (0–5 and 5–10 cm) between both land uses (Figure 1). Adjacent cereal crops considered in these paired comparisons were sampled the same as afforested plots.

### 2.3. Statistical Analyses

A preliminary exploration of differences between the previous land uses (afforestation planted in croplands versus old-fields, N = 43) was performed by two-sample *t*-tests and two-sample Wilcoxon test (non-parametric test (W)) for all the study variables.

Generalized linear models (GLMs) were used to analyze plant diversity indicators in the afforested sites (N = 43) including geographic, climatic, edaphic, geological, and land-use data as explanatory variables of diversity. To start, correlation analysis was performed on quantitative explanatory variables, and highly correlated variables (r > |0.75|) were excluded from subsequent modelling steps. A Poisson error distribution was used to model Abundance, Total_R, and richness by functional group (Annuals_R, Forbs_R, Grasses_R, and Woody_R). A Gaussian error distribution was used to model Total_H. Collinearity and overdispersion were checked for model validation. Variance inflation factors (VIFs) were used to identify collinearity between variables. Colinear variables were removed from the models. Homoscedasticity and normality were evaluated by plotting residuals and predicted values. The best models were selected according to the lower Akaike information criterion (AIC). The proportion of variance explained by the model was calculated as a pseudo $R^2$ = [(null deviance—residual deviance)/null deviance] × 100. Initial model selection was performed following a stepwise procedure. In order to fit the most parsimonious model, any nonsignificant factors were removed from the variables included in the initial model when the AIC was comparable. Finally, the effect of each significant explanatory variable on the response variable was explored by examining factor coefficients and plotting adjusted predictions of the response variable using the most parsimonious fitted model. Comparisons between afforested and adjacent cereal plots were conducted by two-sample *t*-tests. All statistical analyses were performed using R software, and ggplot2, corrplot, car, MASS, and ggeffects packages [57–62].

## 3. Results

### 3.1. Environmental Characteristics and Plant Species in Afforested Sites

Thirty-nine environmental variables were recorded for each of the 43 afforested sites (Table 1; Table S1). A total of 252 plant species were identified (Table S2, taxonomy according to Blanca et al. [63]). Only eight of these had been artificially introduced: *Pinus halepensis*, *Quercus ilex* subsp. *ballota*, *Pinus pinea* L., *Prunus mahaleb* L., *Juniperus phoenicea* L., *Pinus nigra* J.F. Arnold, *Prunus avium* L., and *Quercus faginea* Lam. The remaining native species were divided as 53.57% annuals, 14.29% forbs, 6.35% perennial grasses, and 25.79% woody species. Of these species, a total of 52 (20.63%) belonged to a restricted area of distribution (Table S2): 28 were endemic to the Iberian peninsula and North Africa (11.11%), 15 were endemic to the Iberian peninsula (5.95%), and 13 were endemic to the south and southeast regions of the Iberian peninsula (5.15%). We identified only one endangered species (0.40%) that appears in the Andalucian Red List of threatened species (*Centaurea pulvinata* (Blanca) Blanca). The mean diversity estimated per sample when introduced tree species were included (Total_H) was 2.54 ± 0.40 (SD), and 2.49 ± 0.44 when they were disregarded (H_noaff).

### 3.2. Effect of the Previous Land Use on Plant Diversity and Environmental Variables

Annual_R, CovPhal, D_n_veg, Mg, and Age were significantly higher in afforested sites than in paired cropped sites (t = 4.2305, df = 41, *p*-value < 0.001; Wilcoxon test, W = 319.5, *p*-value < 0.05; W = 305.5, *p*-value < 0.05; W = 320, *p*-value < 0 0.05; and W = 351, *p*-value < 0.01, respectively). Grasses_R, Woody_R, and D_crops were higher in afforested old-fields (W = 74, *p*-value < 0.001; W = 33, *p*-value < 0.0001; and W = 120, *p*-value < 0.01, respectively). No significant differences were found between previous land use for Abun-

dance, Total_R, Total_H, and Forbs_R, and the remaining environmental variables (Supplementary Materials Figures S1–S4).

### 3.3. Effects of Environmental Variables and Soil Properties on Plant Diversity

The quantitative parameters included as explanatory variables in GLMs were selected taking into account a correlation coefficient lower than $|0.75|$ (Figure S5). Additionally, previous land use and lithology were included in the models as categorical explanatory variables. All model results and predicted responses graphs are available in Supplementary Materials.

For Abundance, the most parsimonious model revealed a significant association only with CovPha and Cover (Table S3 and Figure S6).

The selected model for Total_R showed significant negative relationships with D_n_veg, D_shrub, s_K, CSilt, SOC, Age, and Old-field factor (Table 2; Figure 2; Table S4). The model revealed evidence of positive effects of s_Mg, TN, pH, and Cover on Total_R (Table 2 and Figure 2).

**Table 2.** Results for Total_R best and most parsimonious model (pseudo $R^2$ = 75.92%; AIC = 258.3).

|  | Estimate | SE | z Value | Pr(>\|z\|) |
|---|---|---|---|---|
| Intercept | 1.39000 | 0.73410 | 1.894 | 0.058 |
| D_n_veg | −0.00096 | 0.00037 | −2.566 | 0.010 |
| D_shrub | −0.00008 | 0.00004 | −2.091 | 0.037 |
| s_Mg | 0.26380 | 0.09270 | 2.846 | 0.004 |
| s_K | −0.30620 | 0.09801 | −3.124 | 0.002 |
| CSilt | −0.02948 | 0.00944 | −3.121 | 0.002 |
| SOC | −0.21200 | 0.07265 | −2.919 | 0.004 |
| pH | 0.22720 | 0.08642 | 2.629 | 0.009 |
| TN | 3.18600 | 1.17900 | 2.703 | 0.007 |
| Age | −0.06633 | 0.02660 | −2.494 | 0.013 |
| Previous land use Old-field | −0.35190 | 0.09792 | −3.593 | <0.001 |
| Cover | 0.01508 | 0.00289 | 5.219 | <0.001 |

D_n_veg = distance to natural vegetation (m); D_shrub = distance to shrub patches (m); s_Mg = soluble Mg (mg $L^{-1}$); s_K = soluble K (mg $L^{-1}$); CSilt = coarse silt fraction (0.05–0.02 mm, %); SOC = soil organic carbon (%); TN = soil total nitrogen (%); Age = afforestation age (years); Cover = plant cover (%); Previous land use = land use prior to afforestation (old-field or cereal crops).

For functional groups, Annuals_R was significantly negatively associated with D_Qil, K, CSilt, $NO_3$, Sand and Silt+clay lithology, Old-field previous land use, and Age (Table S5 and Figure S7). Positive effects were found for Mg, AW, EC, pH, Siliceous rocks, CovPhal, and Cover.

Significant positive associations were found for pH and Cover on Forbs_R; however, K seemed to exert a negative effect (Table S6 and Figure S8).

For Grasses_R, the selected model showed a positive relationship with AW, CSilt, and Old-field previous land use (Table S7; Figure S9). Evidence of a negative effect of D_shrub and $CaCO_3$ on Grasses_R was found.

The selected model revealed evidence of the negative effects of Gravel and $CaCO_3$ on Woody_R (Table S8; Figure S10). On other hand, a positive effect was found on Woody_R by Old-field as the previous land use.

The most parsimonious model for Total_H included 12 variables, not all associated with significant effects (Table 3; Table S9). D_n_veg, D_shrub, s_K, CaCO3, and Previous_land_use (Old-field factor) had a negative effect on Total_H, whereas s_Mg, pH, CovQil, and Cover showed a positive effect (Figure 3).

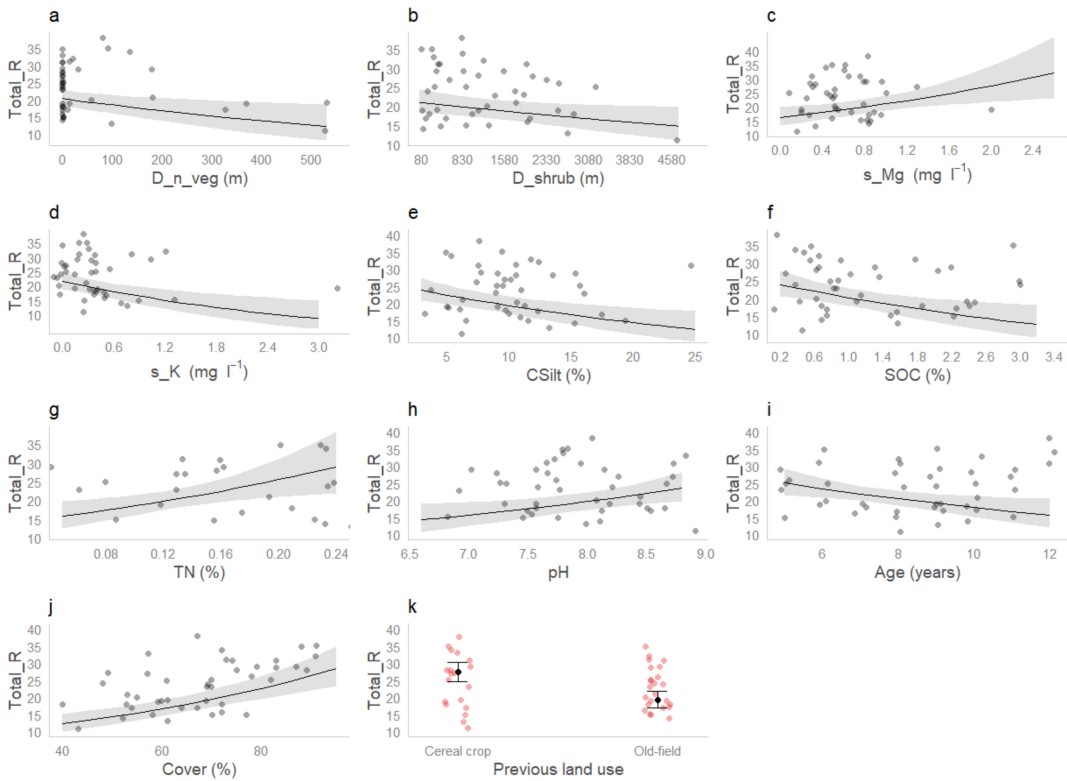

**Figure 2.** Relationships between Total_R and explanatory and response variables identified by the best fit model. Black lines are the fitted lines calculated using the ggpredict function in R, providing the predicted values on the scale of the response. The shaded area represents the 95% confidence interval. Points represent measured values (N = 43). (**a**) D_n_veg = distance to natural vegetation (m); (**b**) D_shrub = distance to shrub patches (m); (**c**) s_Mg = soluble Mg (mg L$^{-1}$); (**d**) s_K = soluble K (mg L$^{-1}$); (**e**) CSilt = coarse silt fraction (0.05–0.02 mm, %); (**f**) SOC = soil organic carbon (%); (**g**) TN = soil total nitrogen (%); (**h**) pH; (**i**) Age = afforestation age (years); (**j**) Cover = plant cover (%); (**k**) land use prior to afforestation.

**Table 3.** Influence of factors identified by the most parsimonious model per plant Shannon Index diversity (Total_H) (pseudo R$^2$ = 74.71%; AIC = 11.24).

| Factor | Estimate | SE | t Value | Pr(> |t|) |
|---|---|---|---|---|
| (Intercept) | −0.92220 | 0.91590 | −1.007 | 0.322 |
| D_n_veg | −0.00098 | 0.00039 | −2.531 | 0.017 |
| D_shrub | −0.00018 | 0.00004 | −4.285 | <0.001 |
| D_Qil | −0.00005 | 0.00003 | −1.969 | 0.058 |
| s_Mg | 0.50330 | 0.13530 | 3.720 | 0.001 |
| s_K | −0.26050 | 0.11160 | −2.335 | 0.026 |
| CSilt | −0.01716 | 0.01152 | −1.490 | 0.147 |
| CaCO3 | −0.00672 | 0.00261 | −2.576 | 0.015 |
| pH | 0.47630 | 0.11550 | 4.123 | <0.001 |
| Age | −0.06244 | 0.03068 | −2.036 | 0.051 |
| Old-field previous land use | −0.35910 | 0.10910 | −3.292 | 0.003 |
| CovQil | 0.13640 | 0.06451 | 2.115 | 0.043 |
| Cover | 0.01312 | 0.00405 | 3.241 | 0.003 |

D_n_veg = distance to natural vegetation (m); D_shrub = distance to shrub patches (m); D_Qil = distance to *Quercus* woodland (m); s_Mg = soluble Mg (mg L$^{-1}$); s_K = soluble K (mg L$^{-1}$); CSilt = coarse silt fraction (0.05–0.02 mm, %); CaCO$_3$ (%); pH; Age = afforestation age (years); CovQil = *Quercus ilex* cover (%); Cover = plant cover (%).

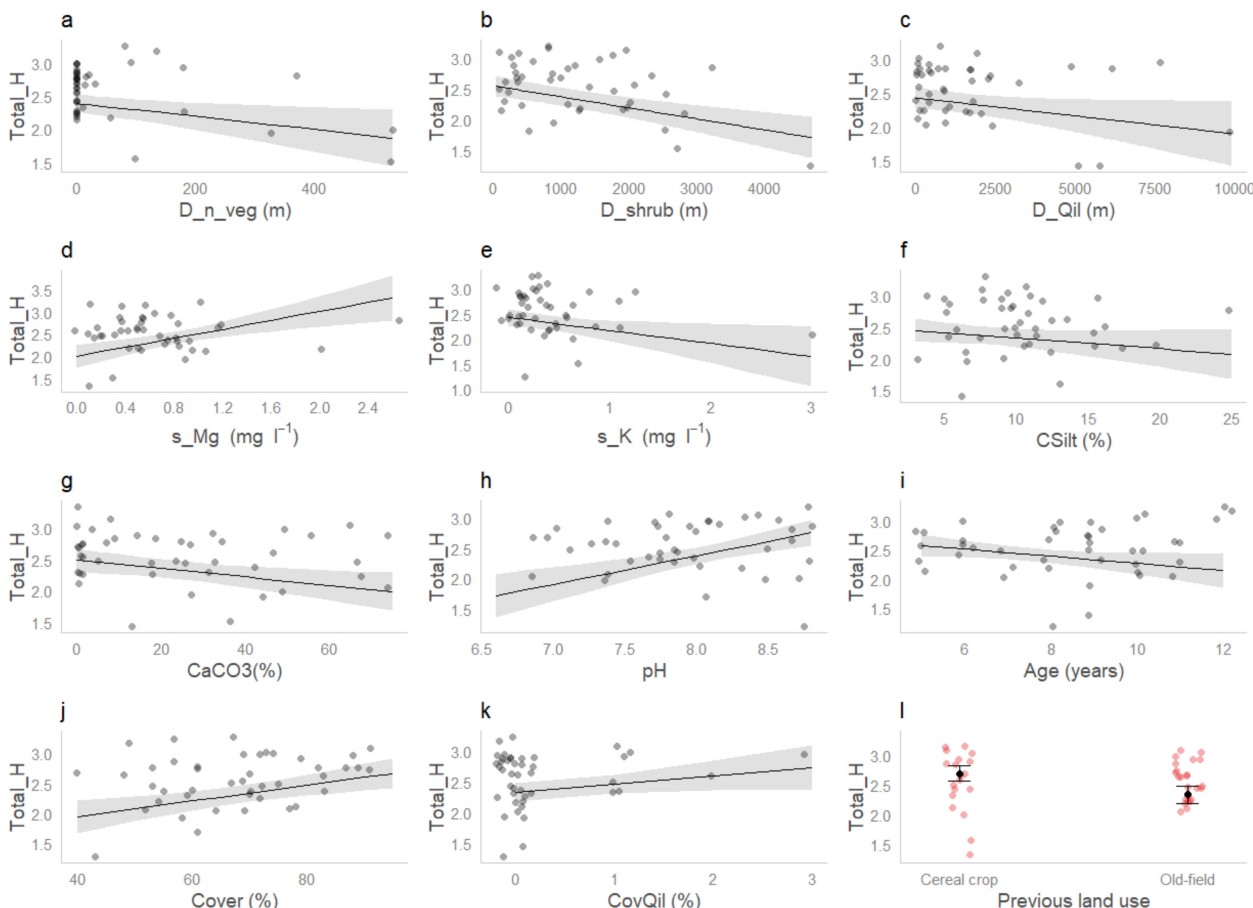

**Figure 3.** Relationship between Total_H and explanatory and response variables identified by the best fit model. Black lines are the fitted lines calculated using the ggpredict function in R, providing the predicted values on the scale of the response. The shaded area represents the 95% confidence interval. Points represent measured values (N = 43). (**a**) D_n_veg = distance to natural vegetation (m); (**b**) D_shrub = distance to shrub patches (m); (**c**) D_Qil = distance to *Quercus* woodland (m); (**d**) s_Mg = soluble Mg (mg $L^{-1}$); (**e**) s_K = soluble K (mg $L^{-1}$); (**f**) CSilt = coarse silt fraction (0.05–0.02 mm, %); (**g**) CaCO$_3$ (%); (**h**) pH; (**i**) Age = afforestation age (years); (**j**) Cover = plant cover (%); (**k**) CovQil = *Quercus ilex* cover (%); (**l**) land use prior to afforestation.

### 3.4. Diversity and Soil Properties Comparison between Afforestation and Cereal Crops

Both Total_R and Total_H were significantly higher in afforested plots than in adjacent paired arable sites (Table 4). Significant differences between land uses were detected in soil properties, mainly within the top 0–5 cm (Table 4). At this depth, afforestation was associated with higher SM33, SM1500, K, SOC (concentration and stock), and TN than cereal crops. At 5–10 cm, most of the variables showed no significant differences, except available P, which was significantly higher in arable plots than in the paired afforested plots.

**Table 4.** Mean (±SD) plant diversity (N = 6) and edaphic properties (N = 18), and paired *t*-test results for land-use comparisons.

|  |  | Paired Sites | | Paired *t*-Test | | |
|---|---|---|---|---|---|---|
|  |  | Cereal Crop | Afforestation | t | df | *p*-Value |
|  | Abundance | 102.50 (18.21) | 97.33 (23.42) | 0.36183 | 5 | >0.05 |
| Plant diversity | Total_R | 12.00 (3.03) | 21.00 (6.20) | −4.0249 | 5 | <0.05 |
|  | Total_H | 1.50 (0.31) | 2.34 (0.63) | −4.9691 | 5 | <0.01 |

**Table 4.** *Cont.*

| | | Paired Sites | | Paired *t*-Test | | |
|---|---|---|---|---|---|---|
| | | Cereal Crop | Afforestation | t | df | *p*-Value |
| | Gravels (%) | 45.31 (15.61) | 46.12 (18.30) | −0.28349 | 17 | >0.05 |
| | SM33 (%) | 19.89 (4.75) | 21.86 (4.84) | −2.4497 | 17 | <0.05 |
| | SM1500 (%) | 12.13 (2.87) | 13.75 (3.30) | −4.1787 | 17 | <0.001 |
| | AW (%) | 7.75 (2.88) | 8.11 (2.39) | −0.71374 | 17 | >0.05 |
| | BD (g cm$^{-3}$) | 0.84 (0.11) | 0.79 (0.14) | 1.9476 | 17 | >0.05 |
| | pH | 8.55 (0.18) | 8.41 (0.27) | 2.4947 | 17 | >0.05 |
| | CaCO$_3$ (%) | 38.84 (17.13) | 34.68 (15.44) | 1.1000 | 17 | >0.05 |
| Edaphic properties | Mg (Cmol$_{(+)}$ kg$^{-1}$) | 1.68 (0.68) | 1.75 (0.94) | −0.28188 | 17 | >0.05 |
| 0–5 cm | K (Cmol$_{(+)}$ kg$^{-1}$) | 0.41 (0.24) | 0.57 (0.23) | −4.1418 | 17 | <0.001 |
| | Na (Cmol$_{(+)}$ kg$^{-1}$) | 0.33 (0.54) | 0.15 (0.20) | 1.2212 | 17 | >0.05 |
| | CEC (Cmol$_{(+)}$ kg$^{-1}$) | 12.83 (5.22) | 14.96 (7.70) | −0.9949 | 17 | >0.05 |
| | P (mg kg$^{-1}$) | 11.55 (6.81) | 15.70 (12.38) | −1.6987 | 17 | >0.05 |
| | SOC (%) | 1.93 (0.96) | 2.75 (1.13) | −3.5696 | 17 | <0.01 |
| | SOC (Mg C ha$^{-1}$) | 7.91 (3.57) | 10.30 (3.56) | −2.9043 | 17 | <0.01 |
| | TN (%) | 0.14 (0.04) | 0.18 (0.05) | −4.539 | 17 | <0.001 |
| | C:N | 13.14 (4.70) | 15.24 (5.07) | −2.0917 | 17 | 0.0518 |
| 5–10 cm | Gravels (%) | 44.92 (15.52) | 44.00 (17.95) | 0.45947 | 17 | >0.05 |
| | SM33 (%) | 20.43 (4.51) | 20.40 (5.06) | 0.02957 | 17 | >0.05 |
| | SM1500 (%) | 12.38 (2.97) | 12.90 (3.17) | −1.2572 | 17 | >0.05 |
| | AW (%) | 8.05 (2.75) | 7.50 (2.49) | 1.0489 | 17 | >0.05 |
| | BD (g cm$^{-3}$) | 0.88 (0.20) | 0.90 (0.17) | −0.90351 | 17 | >0.05 |
| | pH | 8.59 (0.23) | 8.54 (0.24) | 0.77857 | 17 | >0.05 |
| | CaCO$_3$ (%) | 39.29 (17.09) | 33.80 (15.18) | 1.4698 | 17 | >0.05 |
| | Mg (Cmol$_{(+)}$ kg$^{-1}$) | 1.89 (1.09) | 1.54 (0.98) | 1.0955 | 17 | >0.05 |
| | K (Cmol$_{(+)}$ kg$^{-1}$) | 0.39 (0.24) | 0.39 (0.16) | 0.0191 | 17 | >0.05 |
| | Na (Cmol$_{(+)}$ kg$^{-1}$) | 0.14 (0.15) | 0.33 (0.55) | 0.0191 | 17 | >0.05 |
| | CEC (Cmol$_{(+)}$ kg$^{-1}$) | 12.31 (6.11) | 13.03 (6.49) | −0.4215 | 17 | >0.05 |
| | P (mg kg$^{-1}$) | 10.88 (6.59) | 7.19 (5.26) | 2.9986 | 17 | <0.01 |
| | SOC (%) | 1.77 (0.85) | 1.91 (0.91) | −0.74067 | 17 | >0.05 |
| | SOC (Mg C ha$^{-1}$) | 7.42 (3.12) | 7.95 (2.99) | −0.06191 | 17 | >0.05 |
| | TN (%) | 0.14 (0.04) | 0.15 (0.05) | −0.92473 | 17 | >0.05 |
| | C:N | 12.47 (5.47) | 12.82 (4.93) | −0.28237 | 17 | >0.05 |

## 4. Discussion

The influence of afforestation on multifunctionality factors such as plant diversity and soil quality is among the most controversial issues relating to the practice since active restoration has usually been understood as a mere tree plantation [18]. However, recent studies stated that afforested lands do not have to be identified as green deserts [35]. Unlike other agri-environmental schemes, where most species belong to a small set of widely distributed generalist species [64], in our study, we identified almost 25% of species that were either endemic or restricted to a small geographic area. Our estimated indices of diversity are high, similar to numerous Mediterranean plant communities neighboring the study area [65], reinforcing the characterization of this region as a biodiversity hotspot [66].

As expected in young afforested farmland (<12 years old), with little competition for light and soil, species abundance was positively associated with vegetation cover and *P. halepensis* cover. As expected, Total_R and Total_H shared most of the significantly explanatory environmental factors. Lower distance to natural vegetation patches is directly related to new seed sources and dispersal, facilitating colonization by plants, and thus passive (natural) regeneration, especially when plantations are established with native species [67]. In our study area, the more diverse plots in terms of plant species (higher Total_H) were also found closer to established *Quercus* woodlands. The presence of natural or seminatural habitats provides a positive contribution to agroecosystem biodiversity because it increases the local species pool [66,67]. According to Wulf [68], the most efficient

approach to achieving high species diversity in afforested sites is to develop them close to indigenous woodlands, which is consistent with our results.

Surprisingly, elevation and climatic variables had little effect on plant diversity indicators, possibly because of the greater influence of other factors. Moreover, we only detected evidence of sandy soils negatively influencing Annuals_R, although these results should be interpreted with caution due to the small sample size for this lithology type. Similarly, CSilt showed a negative effect on many of the study plant diversity indicators (Total_R, Annual_R, Grasses_R, and Total_H). It is well-known that plant development is driven by nutrient gradients and by soil water availability, which is strongly influenced by soil texture. In this sense, clear positive effects of AW were detected on Annuals_R and Grasses_R.

Contrary to our results, Pausas and Carreras [38] found greater species richness in a carbonate-rich *Pinus sylvestris* L. forest soil, which they ascribed to the positive correlation between $CaCO_3$ and pH rather than higher soil macronutrient availability. This is not the case in our study, where perennial Grasses_R, Woody_R, and Total_H were all negatively influenced by $CaCO_3$. However, in general, pH had a positive effect on Total_R, Annuals_R, Forbs_R, and Total_H in our study.

In contrast to soil Mg content, which is largely derived from soil parental material, soil K forms are associated with SOC [69]. In our study, both SOC and s_K exerted a negative influence on Total_R, while s_K negatively influenced Total_H, and exchangeable K negatively influenced Annual_R and Forbs_R. We might expect a higher SOC to be associated with more established afforested sites. In these older stands, trees are likely to be more competitive, reducing understory species richness. However, this explanation should be taken with caution because our results only cover relatively young afforested plots and no significant interactions or correlation was found between SOC and afforestation age. In any case, the relationships between plant diversity and soil carbon are not clearly understood [39,70]. In addition, our results revealed similar distributions between SOC and K and s_K. High levels of K have been associated with reduced species richness [70,71]. An opposite effect has been observed for soil TN, which may promote species richness in temperate regions [71,72], as we found in our study.

Several authors have provided evidence of links between low soil C:N ratio and greater plant diversity, depending on land use [41,73]. In our case, this tendency was subtle. Although no significant difference was observed, we found lower C:N in afforested former cereal plots (10.96 ± 5.53 SD), where Total_H was higher, than in afforested old fields (11.21 ± 4.33 SD). Nevertheless, the effect of previous land use seemed to be clear on all plant diversity indicators, except for Forbs_R. These results may be related to colonization dynamics and seed sources, which may vary depending on prior land use [17,74]. In young afforested plots, a rapid increase in annual species can be expected after planting trees in agricultural soils due to higher nutrient availability and the mobilization of resources [75]. Additionally, in semiarid afforested old-field sites, soil moisture and canopy density variations can constrain the development of shrubs and herbs [67]. As found in other Mediterranean forests [17], large numbers of annual species in young stands influenced Total_R in our study. Afforested old-field sites were richer in perennial grasses and woody species. These differences, together with the lack of differences in Forbs_R, resulted in lower Total_R and Total_H in afforested old-field sites.

We found that stand age had a significant negative effect on both Total_R and Annuals_R, but a reduced influence upon Total_H. This has been observed before [14,76]. Considering ecological succession, the dominance of annual species immediately after planting is likely to decrease as afforestation matures, with subsequent reductions in Annuals_R, Total_R, and Total_H.

An evaluation of European agricultural schemes failed to show any benefits of afforestation on biodiversity, including plant species, compared to conventional agriculture [27]. However, our results suggested that afforestation may enhance plant diversity when established on land historically used for cereal crops or arable land more generally,

as has been observed previously [6,20,30]. Positive outcomes of afforestation for species richness in semiarid areas were reported, even compared with seminatural habitats such as old-field sites [67]. However, afforestation should be avoided on land used for cereal crops in areas of particular interest for biodiversity or of cultural importance [5], such as preservation of steppe bird habitats or vulnerable or iconic landscapes.

Active restoration initiatives should prioritize the restoration of resilient ecosystem services, especially in drylands. Increasing vegetation cover and plant diversity and improving of soil quality should be considered among the primary objectives [36,37,77]. We included the most frequently used edaphic indicators to assess the success of farm-land afforestation [37]. We observed only limited significant differences in these edaphic variables, possibly because of our short-term assessment. However, those variables that were improved (SM33 and SM1500, both associated with water availability, K, TN, and SOC), are linked strongly to semiarid ecosystem functionality [11,78,79]. Higher SOC and macronutrients in afforested sites relative to arable sites might be related to C inputs to the soil associated with herbaceous biomass production, root turnover, and litterfall from trees [11,80], reflected in the altered C:N ratio in afforested sites. Increases in SOC and TN stocks, and K following afforestation have been confirmed in other studies [20,81,82].

Previous land use is considered to be the main factor influencing P changes following afforestation [83]. Deng et al. [83] reported lower P concentrations following tree planting on previously agricultural land. In our soils, differences between afforestation and paired cereal crop plots were not detected in the top 5 cm, but P concentrations were significantly lower in afforested soils at greater depth (5–10 cm), possibly due to the cessation of fertilizer addition and nutrient uptake by the trees [83]. A comparison between unpaired afforestation and cropland sites in China did not identify any difference in soil total P (0–20 cm) [84].

Our results suggested that afforestation of degraded land with native tree species planted at low densities can be an effective option to increase plant diversity and improve soil quality in the Mediterranean region. However, we encourage policymakers to plan the intervention and its monitoring following technical and territorial criteria before any active restoration to guarantee maximum success in ecosystem service recovery [3].

## 5. Conclusions

We presented evidence concerning the effects of farmland afforestation on biodiversity and soil quality in a semiarid Mediterranean context. The success of restoration activities mainly depended on previous land use; the proximity of woodlands and natural vegetation; soil properties, such as pH or TN; and the afforestation age. Active conversion of arable lands into forests had a positive effect on plant species richness and diversity. Afforestation may act as a reservoir of endemic, rare, or endangered species when low tree densities are used. Some soil properties (SOC, TN, K, and soil moisture) were also improved. These are linked to soil functions such as carbon sequestration, water, and nutrient cycling in Mediterranean areas.

Although our study has some limitations, such as the lack of baseline data before afforestation, limited sample size, and the lack of information regarding cropland manage-ment, we confirmed our hypotheses. For further research, temporal monitoring of changes in plant diversity and soil quality properties, more detailed analysis of the influence of tree density, a larger sample size, and sampling of deeper soil layers are required. Assessment of multiple ecosystem services, for instance, biodiversity metrics of birds and invertebrates, effects of afforestation at the landscape level, and its social and cultural perception would be desirable.

**Supplementary Materials:** The following are available online at https://www.mdpi.com/article/10.3390/f12121730/s1, Table S1: Afforestation locations for plant diversity and environmental variables assessment, Table S2: List of the species identified in this study, Table S3: GLM results for plant abundance (Abundance), Table S4: GLM results for plant total richness (Total_R), Table S5: GLM results for annuals richness (Annuals_R), Table S6: GLM results for forbs richness (Forbs_R), Table S7:

GLM results for perennial grasses richness (Grasses_R), Table S8: GLM results for woody richness (Woody_R), Table S9: GLM results for Shannon diversity index (Total_H), Figures S1–S4: Boxplots showing median, 25th percentile, 75th percentile and range, respectively, in measured afforestation variables and plant diversity indicators, Figure S5: Selected quantitative variables as explanatory variables in GLM, Figure S6. Predicted Abundance responses, Figure S7: Predicted Annuals_R responses, Figure S8: Predicted Forbs_R responses, Figure S9: Predicted Grasses_R responses, Figure S10: Predicted Woody_R responses.

**Author Contributions:** F.B.N. designed and supervised the study; F.B.N., M.N.J. and E.F.-O. collected the data, performed the laboratory analysis, and conceptualized the first version of the manuscript; C.S. analyzed and interpreted the data and wrote the manuscript. All the authors contributed to the results interpretation and the final version of the manuscript. All authors have read and agreed to the published version of the manuscript.

**Funding:** This research was funded by INIA, grant number RTA2005-00009-C02-01.

**Institutional Review Board Statement:** Not applicable.

**Informed Consent Statement:** Not applicable.

**Data Availability Statement:** The data are available from the corresponding author on reasonable request.

**Acknowledgments:** We are grateful to the Delegación Territorial de Agricultura, Ganadería y Pesca en Granada, the Oficinas Comarcales Agrarias, and the Instituto Nacional de Meteorología for providing the farmlands information and meteorological data. We thank F. J. Bonet for sharing geographical data with us. We would like to thank the reviewers for their constructive suggestions for improving our manuscript. Thanks to Andrew Neal for his help in improving the language of the manuscript.

**Conflicts of Interest:** The authors declare no conflict of interest. The funders had no role in the design of the study; in the collection, analyses, or interpretation of data; in the writing of the manuscript; or in the decision to publish the results.

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
