# Peer review of "Effects of Afforestation on Plant Diversity and Soil Quality in Semiarid SE Spain"

_forests, doi:10.3390/f12121730_

Round 1

Reviewer 1 Report

This paper addresses a very important topic on the effect of the many (re)afforestation programs that have been carried out in recent decades, particularly in semi-arid areas, on plant diversity and soil characteristics. It uses 48 afforested sites in SE Spain to explore the relationships between several plant diversity indicators and a large number of environmental variables, such as geographical, climatic or edaphic variables, age, and land-use history. In addition, the authors have compared plant diversity and some soil indicators between paired cereal crops and afforestation, allegedly to assess the success of the intervention (active restoration).  

The authors found that the “restoration effectiveness” mainly depended on the previous land use type, the proximity to natural vegetation, particular soil properties (texture, pH, total nitrogen), and “likely”, the initial tree density and age of the plantation. In addition, afforestation showed higher plant diversity and an improvement of soil parameters (i.e. soil organic carbon, nitrogen, potassium) when compared to cereal crops.

Despite the indisputable importance of the topic addressed, the study is very descriptive, lacks the formulation of concrete hypotheses to be tested, and perhaps that is why it addresses too many predictor variables, which leads to several limitations: too many figures (7, composed of many graphs each) and tables, sometimes with relevant information in supplementary material (e.g. models’ results) , difficulty in interpreting and discussing the results, limited to too general considerations sometimes not supported by the data, list of references too long.

At the same time, the afforestation management that can largely determine plant diversity and soil quality (e.g. shrub clearing) was not considered. All the afforestation had the same management?

Regarding the comparison between reforested areas and cereal crops, it is not clear what the underlying scientific hypothesis is and to what extent this can give an idea of ​​the success of reforestation, since they are distinct land use types with completely different objectives and management methods. It would seem obvious that cereal crops would have lower plant diversity, because the aim is to produce a particular crop(s). With regard to soil, cereal crops are likely to have been fertilized, so the relevance of the comparison is unclear needing a better explanation to make sense, if any.

Furthermore, regarding the methods: i) it is not clear which criteria were used for the selection of sites (the experimental design and a balanced number of samples is essential to reach robust conclusions about the effects of the predictors); ii) data analysis is exploratory, testing and presenting everything and not highlighting the most relevant results, making it harder to discuss them in detail; iii) in the models it is not indicated which N is tested in each case (important because too many predictor variables are used in many of the models), if the predictors were standardized (should have been because they are in different units), or if interactions between predictors were considered.

The study would benefit from being more focused rather than trying to address so many possibilities without concrete hypotheses. Some comments are made below to support these comments and hopefully help the authors to improve the paper.

Also, some details of English sound odd (e.g .too many "the" that are not needed), a review by a native speaker would be advisable.

Abstract

L15 – “since the last decades” sounds odd. Suggestion: over the last decades

L17 – “Plant biodiversity” sounds odd to me, as the concept of biodiversity is much more complex. I would suggest to use “plant diversity”, here and along the paper, since that’s what the author analysed.

L17 – “impacts of afforestation on soil” quality indicators?

L24 –  It is not clear what the authors mean by “restoration effectiveness”, this should be clarified somewhere in the text.

L26 – Why “likely”? This should be clarified or rephrased.

L27 – The term “multifunctionality” should be clearly defined somewhere in the text, or removed.

Introduction

L47 – “the efficiency of the restoration” should be defined.

L61-62 – “most of the studies that have analysed the factors involved in the impacts of active restoration have not included plant biodiversity assessment” [16]. This is supported by only one reference, add more (or some kind of review) or rephrase.

L62-63: “or have not been focused on farmland afforestation or under dry, semiarid or subhumid conditions [17,18]”, but the authors cite 2 examples that were done in semiarid areas. Rephrase

L92 – “(iii) compare plant biodiversity and 92 edaphic properties in paired cereal crops and afforestation sites”. It is not clear why this was done (racionale), nor why this can give an idea of ​​the success of reforestation, since they are distinct land use types with completely different objectives and management methods. It would seem obvious that cereal crops would have less plant diversity, because the aim is to produce a particular crop(s). With regard to soil, cereal crops are likely to have been fertilized, so the relevance of the comparison is unclear.

No concrete hypotheses to be tested are presented, nor what are the expectations and why.

Methods

L107 – Replace by “study area”?

L114-116 – The criteria used for site selection is not clear: “according to their lithology, time since afforestation” is too vague.

L131 – the same, why were these 43 sites selected among all sites?

L136  - Presence?

L138-140 “We estimated the species abundance (number of individuals of each species per plot) and the proportional cover (%) of the aerial parts of each species as well as the bare soil.” Which abundance measure was used in analysis, density or cover?

L147 – i species; “total species abundance in afforestation” or in each plot?

L150 – “Why was conservation status” assessed? It was not mentioned before, and should be.

L183 – “The afforestation administrative expedients”, can you explain what is this?

L184-185: Proportional covers are relative covers? If so, call it relative.

L187-189 – As mentioned before, it is not clear why this was done and which were the expected outcomes, and also it is only N=6, quite low for so many predictors.

L190- 191 – why these depths?

L207-208 - Pseudo R2 ou RL2: [(null deviance - resid. deviance)/null deviance]*100, better call it pseudo R2.

L212 – Use “response variable” instead of variable response

L214 – most parsimonious

Results

General comments:

You have 43 afforestation (N=43?) and test 39 predictor variables at the same time, this is not sound from a statistical point of view. Consider perhaps reducing the number of variables e.g. though PCA to reduce dimensions in main axes of variation.

Along the results the use of variables abbreviations does not help the reader, although I realize you have 39 variables. Too many variables, seems to be the problem.

L222 on – species names in italic

L 224 - Remaining instead of rest?

L236 – W stands for?

L314 – This “case of study” does not make sense to me.

L315-316 – Why should it be lower?

L320 – P higher in cereal crops may be due to fertilization?

Discussion

My general impression is that the discussion touches on relevant points on the topic, but that often these are general considerations that do not result directly from the results of this work. I recognize that it is difficult to approach with the desirable level of depth so many variables analyzed, hence I suggest a more focused approach to data based on concrete questions and hypotheses, studied in turn.

L324 – “multifunctionality” should be explained or defined.

L356 – avoid saying “some authors” when referring to a particular paper. Better “Segura and co-authors found…

L365-367 – This could be clarified by testing for a significant interaction between SOC and afforestation age. Was this tested?

L374-375 – not convincing…

L392-393 – This negative effect of afforestation on plant diversity possibly due to competition by trees, for afforestation with 12 years maximum (relatively young) is not fully convincing. I addition, paragraph L398-401 contradicts what was said before. So, what is the proposed explanation for lower plant diversity?

L415 – The authors do not specify at any point what do they mean by “afforestation effectiveness”.

L430-435 – Very general implications that do not “emerge” from the results of the work.

L441 – No evidence shown about this.

L441 - 444 – Too general as conclusions of this work.

Figures

Legends Fig 2 to Fig 7: represent instead of correspond

Reviewer 2 Report

General Comments

The article carries out a research on the aspect of effects of afforestation on plant diversity and soil quality in semiarid SE Spain. This topic is very useful and relevant in the scientific field.

Specific Comments for Authors

  • Please moderate English changes required.
  • Please prepare high-quality figures.
  • To help improve the information gaps presented in the article, I leave some key references to improve the quality of the article:

Line 69: Hulshof, C. M., & Spasojevic, M. J. (2020). The edaphic control of plant diversity. Global Ecology and Biogeography, 29(10), 1634-1650.

Line 69: Solomou, A. D., Skoufogianni, E., & Danalatos, N. G. (2020). Exploitation of soil properties for controlling herbaceous plant communities in an organic cultivation of lippia citriodora in the mediterranean landscape. Bulgarian Journal of Agricultural Science, 26(1), 79-83.

Line 426: Yang, S., Zhao, W., & Pereira, P. (2020). Determinations of environmental factors on interactive soil properties across different land-use types on the Loess Plateau, China. Science of The Total Environment738, 140270.

-The conclusions could establish the main limitations of the article. In addition, it could also establish future lines of research that are open or pending after the research is carried out.

I look forward to receiving the revised MS.

Best regards

Round 2

Reviewer 2 Report

The paper is accepted without any further changes.

Best regards

Author Response

Dear Reviewer,

We have had the help of one of our native English colleagues to do a deeper English review.

Many thanks for your helpful feedback,

The authors.